# The Progress in the Application of Dissolving Microneedles in Biomedicine

**DOI:** 10.3390/polym15204059

**Published:** 2023-10-12

**Authors:** Xueqing Yu, Jing Zhao, Daidi Fan

**Affiliations:** 1Engineering Research Center of Western Resource Innovation Medicine Green Manufacturing, Ministry of Education, School of Chemical Engineering, Northwest University, Xi’an 710069, China; 2Shaanxi Key Laboratory of Degradable Biomedical Materials and Shaanxi R&D Center of Biomaterials and Fermentation Engineering, School of Chemical Engineering, Northwest University, Xi’an 710069, China; 3Biotech & Biomed Research Institute, Northwest University, Xi’an 710069, China

**Keywords:** dissolving microneedles, drug delivery, cancer therapy, wound healing, diagnostic, cutaneous disease

## Abstract

In recent years, microneedle technology has been widely used for the transdermal delivery of substances, showing improvements in drug delivery effects with the advantages of minimally invasive, painless, and convenient operation. With the development of nano- and electrochemical technology, different types of microneedles are increasingly being used in other biomedical fields. Recent research progress shows that dissolving microneedles have achieved remarkable results in the fields of dermatological treatment, disease diagnosis and monitoring, and vaccine delivery, and they have a wide range of application prospects in various biomedical fields, showing their great potential as a form of clinical treatment. This review mainly focuses on dissolving microneedles, summarizing the latest research progress in various biomedical fields, providing inspiration for the subsequent intelligent and commercial development of dissolving microneedles, and providing better solutions for clinical treatment.

## 1. Introduction

Subcutaneous injection is one of the common drug administration methods in clinics. However, it can be painful and invasive, can produce sharp and biodangerous waste, and needs to be performed [1,2] by trained healthcare personnel [3]. In recent years, the strategy of transdermal drug delivery has become an important therapeutic option to complement oral, subcutaneous, and intravenous injections [4,5]. Transdermal administration avoids first-pass liver metabolism compared to subcutaneous administration; it is painless compared to injection; and it is a needle-free device that avoids the risk of disease transmission associated with the re-use of needles, thus reducing the unsafe factors commonly caused by medical waste, especially in developing countries [6,7]. However, the stratum corneum (SC), a layer of skin formed by dead corner cells and located in the outermost layer of the skin, with a thickness of 10–15 μm, is the main barrier to transdermal drug delivery, and can severely reduce the efficiency of drug delivery and limit the types of drugs that can be delivered transdermally [8,9,10].

In 1976, a new technology called microneedles was first introduced to overcome the limitations of traditional transdermal drug delivery [11]. MNs consist of micron-scale needle arrays with heights of approximately 50–900 μm. MNs can be manufactured with microfabrication techniques using a variety of materials and geometrics, because they disrupt the SC and epidermal layer and form microscale drug delivery channels without touching the nerve fibers or blood vessels located in the dermis of the epidermis and epidermal layer. Microneedle technology can significantly improve the efficiency of drug delivery [12,13]. It is also possible to increase the types of drugs transported in a painless and minimally invasive manner [14]. As a result, the device is easy to use and painless compared to traditional invasive injection and/or oral strategies with functional advantages [15,16]. MNs can not only help drug molecules bypass first-pass metabolism and gastrointestinal degradation, but also broaden the application range of their drug types, regardless of their molecular weight or hydrophilicity [17,18]. There are currently no confirmed reports that MNs cause or increase the chance of skin infections [19], nor that they affect normal skin function [20]. Numerous preclinical studies and a limited number of clinical trials have now shown that MNs can be used to deliver DNA, vaccines, insulin, and human growth hormone. In addition, MNs have been extensively studied for blood sampling, signal monitoring, and biosensors. This means that MNs have a broad market for transdermal drug delivery, vaccine preparation and biologics. Large amounts of MN have entered clinical trials for the treatment of various diseases, showing its universal effectiveness [12,21,22].


In general, as shown in Figure 1A [23], microneedles can be divided into solid microneedles, coated microneedles, hollow microneedles, and dissolved microneedles [24]. Different kinds of microneedles can deliver drugs through different mechanisms.

Dissolving or swelling microneedles can encapsulate drugs in a polymer matrix and completely dissolve or swell after insertion into the skin [25]. Biodegradable polymer MN systems are typically manufactured by encapsulating drugs or embedded micro/nanoparticle particles into degradable matrix materials. In contrast to coated microneedles, polymer microneedles can be completely dissolved in the skin, so no bio-hazardous sharp object waste is left behind after use [26,27]. These microneedles are typically made only from safe, inert, water-dissolving materials, such as dissolving polymers and sugars. Dissolving microneedles are manufactured primarily using micromolds filled with solvent casting (with water as the usual solvent), as well as filled polymers cured in the mold and polymerized in situ (Figure 1B) [28]. Currently, a variety of materials, including carboxymethyl chitosan (CMC) [29], chondroitin sulfate [30], glucan [31], dextrin, polyvinylpyrrolidone (PVP) [32], polyvinyl alcohol (PVA) [33], polylactic acid-glycolic acid copolymer (PLGA), fibroin protein [34], etc., have been used as microneedles by dissolving them in water. The microneedles are obtained by filling the mold cavity and drying it, sometimes with the additional use of vacuum and/or centrifugal force. In addition, polymer-formed hydrogels, such as N-vinylpyrrolidone and/or methacrylic acid [35], which are added to the mold as liquid monomers and cross-linked under ultraviolet radiation, are also types of dissolving microneedles. In contrast to these highly water-dissolving, rapidly dissolving microneedles, cross-linked microneedles can achieve slow drug release through slow biodegradation of the polymer in the skin [36,37].
Figure 1(**A**). Schematic diagram of different types of microneedles applied to (a) skin and (b) drug delivery [23]. (**B**). Relevant content of dissolution microneedles [28]. Reproduced with permission from the Journal of Controlled Release and International Journal of Pharmaceutics, respectively.
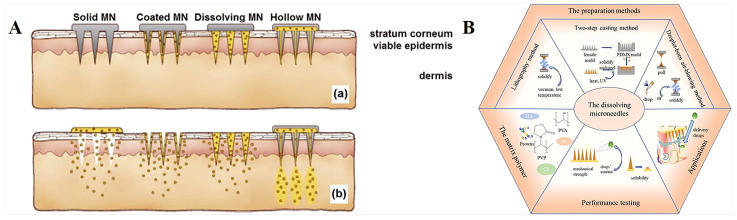



Over the past few decades, dissolving microneedles have been used in various fields, including anti-cancer, dermatology treatment, vaccine delivery, insulin delivery, and biomarker detection. Furthermore, in addition to single drug delivery, the design of microneedles allows for customizable special structures and intelligent response systems, such as controlled drug release; multi-therapy; and targeted delivery to specific sites, such as the heart [36], blood vessels [38], brain [39], etc. In this review, we focus on the latest designs of and research on dissolving microneedles in biomedicine-related fields, such as wound healing, anti-cancer, skin diseases, biomarker detection, diabetes treatment, and other fields; discuss and outline the therapeutic strategies and advantages of microneedles as therapeutic tools, hoping to provide more application ideas for dissolving microneedles; and provide prospective guidance for future clinical transformation. This review is more inclined to introduce the design highlights of microneedles in different diseases and different fields in recent years and to pay more attention to the initial design of microneedles, aiming to provide researchers with more ideas for the subsequent development of different kinds of microneedles.

### Data Collection

Numerous literature searches have been conducted to clarify the applications of dissolvable microneedles in various fields of biomedicine. Firstly, the current research data on soluble microneedles were collected by searching X-mol. https://www.x-mol.com/ (accessed on 6 July 2023). The following search criteria were used: “dissolving microneedle”. The time frame was limited to after 2000 (the search date was May 2023). The search criteria returned 700 results. In addition, based on our chosen application direction, further searches were conducted using keywords such as “cancer; microneedle”. The time range was limited to after 2000, and the search condition returned 295 results. “Wound; microneedle”, with a time frame limited to after 2000, returns 237 results under search conditions; “diagnostic; monitor; microneedle”, with a time frame limited to after 2000, returned 53 results. According to different application directions, some representative articles were selected to complete this review.

## 2. Dissolving Microneedle for Cancer Therapy

As one of the main diseases threatening human health, the treatment of cancer has been the focus of attention all over the world. In addition to commonly used chemotherapy drugs, with the development of nanotechnology, more and more treatment methods are constantly improving the survival rates of cancer patients and reducing the recurrence of cancer. In addition, the on-demand delivery and responsive release of photothermal and photodynamic, gene, and immunotherapy drugs can also be achieved through microneedles. In the past decade, MN has been widely used for the transdermal delivery of anticancer drugs, showing improved drug utilization, reduced toxic side effects, enhanced tumor targeting ability, and low off-target toxicity [40].

Initially, microneedles were used as delivery tools for some first-line anticancer drugs to improve drug availability and target cancer cells [41,42]. Huang et al. [43] prepared the matrix material for the microneedles, Dextran methacrylate (DexMA), by grafting methacrylic acid onto Dextran (Figure 2A). The crosslinking degree of DexMA and the mechanical strength of microneedles was controlled through photocrosslinking so that the prepared MNs were able to successfully penetrate the epidermal layer and achieve sustained drug release. The anticancer drugs doxorubicin (DOX) and trimetinib (Tra), approved by the Food and Drug Administration, were loaded into the microneedle system at the same time. In addition to its anticancer effect, Tra can also reverse P-gp-mediated multidrug resistance (MDR) and effectively block P-gp’s efflux of DOX. The results of B16 xenograft therapy in nude mice have shown that the DOX+Tra-MNs group has superior anti-tumor effects compared to conventional administration (injection and oral).

Photothermal therapy is also often combined with microneedles for the treatment of cancer [44]. Wang et al. [45] fabricated a novel near-infrared (NIR) light-triggered separable microneedle system to supply controlled chemo-thermal therapy to superficial tumors by co-loading a self-developed NIR-II fluorescence fluorophore Flav7, as well as the antitumor drug doxorubicin (DOX), in phase-change polycaprolactone (PCL) arrowheads on a dissolving base (Figure 2B). Flav7 provides microneedles with near-infrared light-triggered photothermal and drug release properties, as well as IR-II imaging capabilities, to guide chemotherapy–thermal combination therapy. It is a dissolving support base that ensures that the needle is embedded in the skin after insertion and imparts MNs with separation capabilities to reduce their contact time with the skin. The microneedle combines chemotherapy and photothermal therapy, and also has the function of NIR-II fluorescence imaging to guide chemotherapy and hyperthermia. Chen et al. [46] developed a light-activated microneedle (MN) system that can repeat and simultaneously deliver photothermal therapy and chemotherapy to superficial tumors, and play a synergistic anticancer role (Figure 2C). The upper backing part of the system is composed of PVA/PVP, which plays a supporting role and quickly dissolves after insertion into the skin. The tip part is composed of polycaprolactone loaded with photosensitive nanomaterials (lanthanum hexaboride) and anticancer drugs (doxorubicin; DOX). When exposed to near-infrared light, LaB6 absorbs laser energy and converts it into heat, inducing MN melting at 50 °C, which triggers MN to release DOX to the tumor. This photoactivated MN with a unique embedding function provides an effective, user-friendly, and low-toxicity option for patients requiring long-term or multiple cancer treatments.

In addition to photothermal therapy, in recent years, combining the microneedle system with photodynamic therapy to allow for greater uniformity and depth in photosensitizer delivery to tumor sites, thus providing better treatment outcomes and an easier pathway, offers a promising strategy for the clinical application of PDT [47,48]. Tham et al. [49] developed a mesoporous nanocarrier with dual loading of photosensitizers and clinically relevant drugs for combination therapy while utilizing microneedle technology to promote its penetration into deep skin tissues. Skin fluorescence imaging shows that microneedles can encourage the nanocarrier to penetrate the skin epidermis and reach deep melanoma sites. The combination of PDT and targeted therapy with nanocarriers has been proven to have a superior therapeutic effect in the xenotransplantation of melanoma mice.

Immunotherapy using MNs has also shown countless prospects, especially in overcoming skin barriers that hinder the percutaneous delivery of large molecules such as antigens, antibodies, and DNA [50,51,52]. Dosta et al. [53] developed a microneedle (MN) platform that can simultaneously deliver immune activators and collect interstitial skin fluid (ISF) to monitor treatment responses (Figure 2D). MNs are synthesized from cross-linked hyaluronic acid (HA) and loaded with drug CpG oligonucleotide (TLR9 agonist) containing model immune regulation nanoparticles for cancer treatment in melanoma and cancer models. After the patch is removed and digested, the treatment response is monitored through longitudinal analysis of the immune cells embedded in MNs.
Figure 2(**A**). Schematic of DexMA hydrogel MNs that can be used for continuous transdermal release of drugs [43]. (**B**). Flav7 + DOX co-loaded separable MNs for light-triggered chemo-thermal therapy of SMTs [45]. (**C**). Schematic illustrations of a combination of chemotherapy and photothermal therapy using near-infrared (NIR) light-activatable microneedles (MNs) [46]. (**D**). Representative scheme of a hyaluronic acid (HA)-based microneedle platform for the delivery of immunomodulatory drugs (CpG-ODNs), complexed with poly (beta-amino esters) (PBAEs) and simultaneous sampling of interstitial fluid (ISF) for the recovery of immune cells ex vivo [53]. Reproduced with permission from the Carbohydrate Polymers, Chemical Engineering Journal, ACS Nano, and Theranostics, respectively.
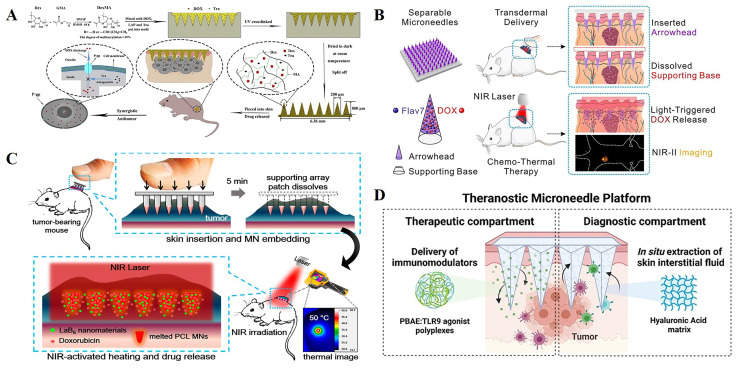



The use of MN broadens the delivery mode of cancer therapy, with various hydrophilic and hydrophobic drugs being able to penetrate the skin and reach the lesion area through microneedles, greatly improving drug utilization. In addition, MNs improve drug stability, making synergies of multiple therapies possible and reducing the need for trained operators, thus making them promising tools.

## 3. Dissolving Microneedles in Wound Healing

As the largest organ of the human body, the skin is easily damaged by the surrounding environment [54]. Bacterial infections, diabetic wounds, bedsores, burns, and other skin lesions pose a serious threat to people’s lives and health. Wound healing goes through a series of roughly continuous, but overlapping, stages: hemostasis, inflammation, proliferation, and remodeling [55]. Each stage is crucial to the final healing of the wound. Microneedle arrays can be used to improve delivery efficiency, thereby improving healing. MN improves the efficiency of transport by bypassing various physical and chemical barriers to deliver treatment to the target area with an improved spatial distribution [56]. In chronic wounds, the presence of eschar, exudate discharge, and harsh chemical microenvironments rich in various enzymes can undermine the effectiveness of local drug delivery therapies [57,58]. The MN system can increase the availability of various therapeutic agents by controlling the drug content of a single needle in a controlled spatial distribution [59,60,61].

For infected wounds, the priority in the wound healing process is to kill bacteria and eliminate inflammation [62,63,64]. Local or percutaneous administration in the form of antibacterial cream, ointment, gel, or lotion containing antibacterial agents is the preferred method for the treatment of chronic skin infections [65]. Combining these with microneedles can significantly improve the antibacterial efficacy and reduce off-target toxicity [66,67,68,69]. Chi et al. [70] prepared a microneedle array patch (CSMNA) (Figure 3A). Chitosan, which has natural antibacterial properties, endows microneedles with antibacterial properties, preventing bacterial infection in the early stages of wound repair. At the same time, vascular endothelial growth factor (VEGF) was wrapped in the pores of CSMNA with thermosensitive hydrogel to promote wound healing by promoting angiogenesis during the remodeling period of wound repair. It is worth noting that in infected wounds, especially chronic wounds, the presence of biofilms greatly weakens the therapeutic effect of antibacterial substances such as antibiotics. Deng et al. [71] developed a dual-function directed antibacterial sericin microneedle (OASM) (Figure 3B). OASM has a high antibacterial performance; it loads zinc oxide nanoparticles onto the tip of the needle as antibacterial modules. Although these applied antimicrobials can play a certain role in combating microbial plankton infections, most of them have difficulty achieving the purpose of biofilm elimination [72]. This may be due to the inefficiency of the antimicrobial agent in penetrating the biofilm through the EPS barrier. The net negative charge of EPS sequesters positively charged antimicrobials or repels negatively charged antimicrobials. MN patches can directly penetrate biofilm barriers or be used to deliver antimicrobials to target sites. By directly delivering zinc oxide nanoparticles to the interior of the wound through microneedles, the biofilm barrier is broken, and the antibacterial effect of zinc oxide nanoparticles can be greatly improved. In addition, sericin, as a component of microneedle materials, significantly promotes skin repair through hair follicle regeneration and angiogenesis. Therefore, combining an antimicrobial agent that can kill microorganisms with an agent that destroys the biofilm barrier structure would be an ideal therapy for eliminating multi-microbial biofilms [73,74,75,76].

Su et al. [79] reported a Janus-type antibacterial dressing, which was composed of an electrospun nanofiber membrane and dissolving microneedle array and was able to effectively deliver antimicrobial peptides inside and outside the biofilm. The microneedle dressing was able to eradicate MRSA biofilms in human skin wound infection models in vitro and in type II diabetes mouse wound infection models. Xu et al. [80] loaded chloramphenicol (CAM)- and gelatinase-sensitive gelatin nanoparticles into dissolving microneedles and created patches (CAM@GNPs). Microneedles can respond to gelatinase produced by active bacterial community, and can dissolve and evenly release CAM@GNPs. Compared to direct administration of CAM, microneedles exhibit less off-target toxicity, which is beneficial for wound healing.

During other stages of wound healing, growth factors, immunomodulators, etc., can also be delivered to the wound through microneedles to promote rapid wound healing [81,82,83]. Xu et al. [77] proposed a new live microneedle (MN) patch for the local delivery of bioactive platelet-derived growth factor D (PDGF-D) and human adipose-derived stem cells (ADSCs) to treat diabetic foot wounds (Figure 3C). ADSCs, along with PDGF-D, were delivered to a relatively ductile wound bed with minimal tissue damage by encasing them within semisolid MNs. In addition to its role in the proliferation, differentiation, and migration of ADSCs, PDGF-D integration can enhance ADSC cell function, further facilitating wound healing in the DU. The use of MN increases the closure rate of diabetic wounds, improves re-epithelialization, and increases angiogenesis in animals treated with it. Ma et al. [78] proposed a microneedle patch with a core–shell structure (Fe-MSC-NVs/PDA-MN) (Figure 3D). Ferrum–mesenchymal stem-cell-derived artificial nanovesicles (Fe-MSC-NVs) containing multifunctional therapeutic cytokines were encapsulated in the inner core of the hyaluronic acid (HA) tip in order to accelerate angiogenesis. The PDA NPs were encapsulated in the outer methacrylate hyaluronic acid (HAMA) shell of the MN tips to overcome the adverse impacts from reactive oxygen species (ROS)-derived oxidative stress. These antioxidant, anti-inflammatory, and pro-angiogenic properties have shown promising results for diabetic wound healing.

As a new type of transdermal drug delivery method, microneedles can break down barriers at the wound site; improve drug delivery efficiency; and achieve various antibacterial, proliferative, and angiogenesis-related effects to improve wound healing.

## 4. Dissolving Microneedles for Diagnostics and Monitoring

Although MNs were originally designed to be developed for the transdermal delivery of drugs and vaccines, their suitable size is large enough to penetrate the cuticle of the skin and enter the dermal ISF without triggering pain-sensing neurons deep in the skin. Thus, microneedles are also a technology whose properties are well-suited for direct access to the dermal ISF in a minimally invasive manner. They provide an excellent platform for transdermal diagnosis and monitoring [84]. In addition to their ability to cross the stratum corneum, MNs’ ability to come into direct contact with the dermal ISF provides an opportunity to sample fluids for external analysis or for directly measuring physiological parameters within the skin. MN-based medical sensing technologies can be divided into two categories. The first is electrochemical biosensors; generally, solid and hollow MNs are used as electrode substrates for further modification [85,86,87,88]. The other is the direct extraction of interstitial fluid (ISF) by means of polymer MNs or hydrogel MNs. In an MN-based electrochemical biosensor, the concentration of the target sample (glucose, lactic acid, alcohol, urea, amino acids, therapeutic drugs, etc.) reflecting the human condition is converted into an electrical signal through the MN electrode [89]. In MN-based direct ISF extraction, because ISF exists in the skin in large quantities, when hydrogel MNs with three-dimensional cross-linked network structures penetrate the skin, ISF is absorbed to form a swollen state [90]. By extracting ISF or by real-time detection, researchers can analyze these therapeutic drugs, proteins, and ions that reflect the physiological conditions of the body.

For electrochemical biosensors, Liu et al. [91] used a microfabrication process and an electroplating process to fabricate the electrochemical sensing electrodes on the microneedles. Then, glucose oxidase (GOD) was immobilized on the working electrode of the sensor. After the microneedles were inserted into the dermis layer of mouse skin, in the presence of subcutaneous glucose, H_2_O_2_ was produced through an enzymatic reaction on the working electrode to generate a current signal response. More specifically, glucose reacted with GOD to produce H_2_O_2_, which diffused to the surface of the working electrode and then oxidized to produce a current signal. The microneedle biosensor was constructed with a two-electrode configuration, including a Prussian blue-coated Au working electrode and an Ag/AgCl counter/reference electrode for the detection of H_2_O_2_, and the current response increased linearly with the increase in H_2_O_2_ concentration, ranging from 0.8 mM to 36 mM. The subcutaneous glucose levels were monitored continuously and in real time. For the purpose of extracting ISF, Chang et al. [92] developed a swelling MN patch able to quickly extract ISF (Figure 4A). The MN patch was made of methacrylic acid hyaluronic acid (MeHA), and was further cross-linked by ultraviolet radiation. Due to the high hydrophilicity of MeHA, this MN patch was able to extract sufficient ISF in a short period without the need for additional equipment assistance. The extracted ISF metabolites could be effectively recovered from MN patches through centrifugation for the subsequent offline analysis of metabolites such as glucose and cholesterol. Real-time detection of blood glucose concentration and timely administration of insulin are very important for diabetic patients. At present, many studies have combined microneedles with blood glucose detection and even responsive insulin release so that patients with diabetes can monitor the indicators at any time and take their medication at the correct time [93,94,95,96]. Li et al. [97] described a fluorescence-amplified origami microneedle (FAOM) device that integrated tubular DNA origami nanostructures and glucose oxidase molecules into its internal network for the quantitative monitoring of blood glucose (Figure 4B). The glucose molecules collected by microneedles were converted into proton signals through the catalysis of oxidase. The proton-driven mechanical reconstruction of DNA origami tubes separated fluorescent molecules from their quenchers, ultimately amplifying glucose-related fluorescence signals. For the integration of tubular DNA origami nanostructures and glucose oxidase molecules, a rectangular DNA origami sheet was first prepared by assembling an M13 bacteriophage genome DNA strand with multiple staple strands. Subsequently, the rectangular sheet was locked into a hollow tubular DNA origami sheet with eight pairs of proton-sensitive DNA fasteners. The black hole 3-quencher (BHQ3) and Cy5-labeled DNA fasteners served as switchable fluorescent beacons, in which proton-sensitive sequences were hybridized and shielded in DNA duplex regions. In the locked state, the blackhole quencher enabled the adjacent fluorescent molecule signal to switch off. The tubular DNA structures in the turn-off state and glucose oxidase molecules were physically trapped in the methacryloyl hyaluronic acid (mHA)-based inner-core network. When attached to the skin tissue, the FAOM device was able to collect the glucose-containing interstitial fluid in situ and transfer the glucose stimuli into proton signals catalyzed by glucose oxidases. Chen et al. [98] prepared an insulin-loaded microneedle patch by semi-interpenetrating a borate-containing hydrogel with biocompatible silk fibroin (Figure 4C). The presence of borate hydrogel enabled insulin to attain the characteristics of glucose-responsive diffusion release, and crystalline silk fibroin components were used as matrix enhancers to verify skin permeability. The use of microneedle technology makes monitoring methods more stable, safe, cost-effective, and able to provide acute and continuous blood glucose control with minimal dependence on patient compliance.

In addition to detecting blood sugar and providing insulin as needed, microneedles are also used to detect DNA and RNA [99,100,101,102], cytokines [103,104,105,106], exosomes [107], small molecules [108,109,110], etc. By combining them with electrochemistry, the monitoring of movement status and wounds can also be achieved [111,112,113]. Guo et al. [112], inspired by the structure of shark teeth, designed a microneedle patch for intelligent wound management (Figure 4D). The microfluidic channel was composed of a microneedle array and a porous ordered structure, and enabled the microneedle patch to analyze various inflammatory factors. In addition, MXene electronic products were patterned on microneedle patches to achieve sensitive motion monitoring. In addition, the use of drug-loaded biomimetic microneedle patches promoted the recovery of full-thickness skin wounds. Xu et al. [103,104,105,106] designed functional carbon nanotube biointerface-based wearable MNs for the minimal invasive capture and real-time monitoring of inflammatory cytokines in ISF with high sensitivity, specificity, and stability. This wearable MN sensor used a CNTs-chitosan membrane as the conductive channel and a cytokine-specific antibody as the capturer to monitor the cytokine storm and the inflammatory response to perform real-time monitoring of cytokines via electrochemical analysis in situ.
Figure 4(**A**) (a) Schematic representation of the rapid extraction of ISF by crosslinked MeHA-MN patches. (i) The crosslinked MeHA network within MN patch is dried and compressed. (ii) In skin, the MeHA-MN patch rapidly swells and extracts ISF that contains metabolites. (iii) After removal, MN patch retains the structure integrity. (iv) Extracted ISF and metabolites can be efficiently recovered from MN patch by centrifugation for subsequent offline analysis. (b) Schematic of the fabrication process of a crosslinked MeHA-MN patch. (c) Scanning electron microscopy (SEM) image of a crosslinked MeHA-MN patch; inlet is a false-color image (scale bar: 500 µm). (d) Optical image of a crosslinked MeHA-MN patch (scale bar: 1000 µm) [92]. (**B**) Schematic illustration of a fluorescence-amplified origami microneedle (FAOM) device for quantitatively monitoring blood glucose [97]. (**C**) (a) Glucose-dependent equilibria of PBA derivatives. (b) Schematic representation of the formation of semi-IPN hydrogel. (c) “Skin-layer” controlled glucose-responsive insulin release from the MN array patch [98]. (**D**) Schematic diagram of shark-tooth-inspired microneedle dressing for intelligent wound management including motion sensing, biochemical analysis, and healing [112]. Reproduced with permission from Advanced Materials, Advanced Functional Materials, and Acs Nano, respectively.
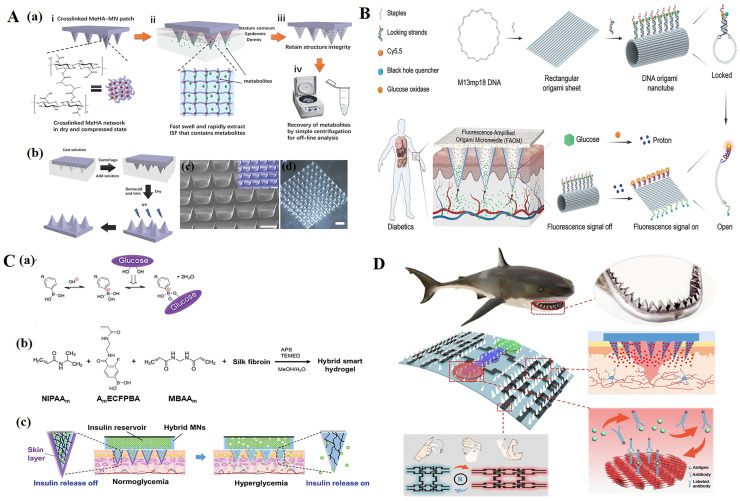



Therefore, with MN-assisted monitoring of technology or extraction of ISF, healthy people or patients without professional training can self-diagnose, simplify the monitoring process, and avoid the potential dangers and problems caused by delayed medical guidance. Combined with MN assistive technology, more information regarding physical conditions can be obtained in a minimally invasive, fast, and convenient way.

## 5. Dissolving Microneedle for Cutaneous Disease

As a global public health problem, skin diseases seriously affect the quality of life of patients [114]. The presence of the stratum corneum (SC) of the skin also severely impedes the transdermal penetration of drugs, making it extremely difficult for them to cross the skin [115]. Microneedles can increase skin permeability, increase drug concentration in local skin lesions, and reduce systemic toxicity. Microneedles have been used to treat a variety of common skin conditions, such as acne, hair loss [116,117,118], atopic dermatitis (AD) [119,120,121,122,123], psoriasis [124,125,126,127], and scarring [128,129,130,131,132,133,134].

Wu et al. [135] encapsulated the biologics IL-17mAbs and MXene in a microneedle patch for psoriasis treatment (Figure 5A). The microneedle patch was able to quickly dissolve and release its load into the dermis under near-infrared light irradiation, producing an effective anti-inflammatory effect around psoriatic lesions and effectively reducing psoriatic-like skin inflammation. Wu et al. [136] developed a microneedle system for the local treatment of hypertrophic scars (Figure 5B). A cyclodextrin metal–organic framework (CDF) containing quercetin (QUE) was coated with HSF membrane (QUE@HSF/CDF), then dispersed in Bletilla striata polysaccharide (BSP)-fabricated dissolvable microneedles (BSP-MNs-QUE@HSF/CDF) for local administration. This regulated Wnt/β- catenin, and the JAK2/STAT3 pathways reduced the expression of collagen I and III in HS and improved the therapeutic effect on HS.

Chiu et al. [137] developed a microneedle (MN) formulation based on poly-γ-glutamate (γ-PGA) that not only protected EGCG from oxidation, but also acted as an immunomodulator to downregulate the helper T cell type 2 (Th2) immune response, effectively delivering EGCG into the skin to improve atopic dermatitis (AD) symptoms (Figure 5C). A weekly MN dosing regimen may provide patients with a more convenient, treatment-equivalent option than daily topical dosing, and may improve adherence and long-term continuity in AD therapy. In addition, Xiang et al. [138] used sodium hyaluronate as a microneedle material to encapsulate nanoparticles formed by a zinc porphyrin-based metal–organic framework and zinc oxide (ZnTCPP@ZnO) to prepare a microneedle patch for the treatment of acne (Figure 5D). With 15 min of ultrasound irradiation, the antibacterial efficiency of Pseudomonas acne was 99.73%. This would reduce the levels of acne-related factors, promote the proliferation of fibroblasts, and thus promote skin repair.

In addition to the aforementioned applications, microneedles are currently widely used in the treatment of other diseases, such as long-term contraception (Figure 6A) [37,139,140,141], rheumatoid arthritis (Figure 6B) [142,143,144,145,146,147], weight loss (Figure 6C) [148,149,150,151], Alzheimer’s disease [152,153,154], anesthesia [155,156,157,158], myocardial infarction (Figure 6D) [159,160,161,162], vaccine delivery [163,164,165,166], etc. We believe that microneedle technology will be applied in more fields in the future to improve treatment efficiency for diseases and greatly facilitate patient use.

## 6. Conclusions

In the decades since the use of MNs as drug delivery systems, significant progress has been made in the field of transdermal drug delivery. MNs can significantly enhance drug penetration through the skin. The microneedles mentioned in this paper are not only used as simple drug delivery tools. Through the recent introduction of microneedles in the field of biomedicine in this paper, we hope that the treatment status and efficiency of various diseases will be improved due to the design of microneedle morphology, the selection and modification of its materials, and the interaction between microneedle materials and drugs. In addition, in this paper, we mentioned the application of dissolving microneedles in many biomedical fields, and we hope that microneedle technology can be more widely used in other aspects. However, to widely apply MNs in various fields, their relatively low cost and simple operation are still worth considering. Most MNs designed for therapeutic use are still in the development stage, and how to ensure the uniformity of microneedle products and the approved use of microneedle materials requires more extensive efforts and strict supervision. Once these issues are resolved, we optimistically expect that MNs will become a superior alternative to the current management methods.

## Figures and Tables

**Figure 3 polymers-15-04059-f003:**
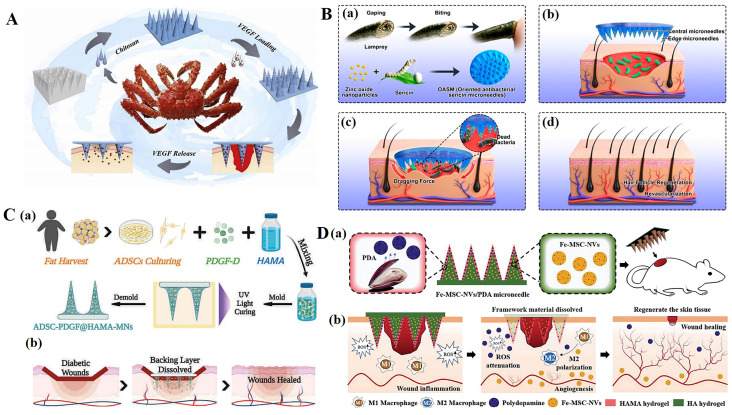
(**A**). Scheme of the fabrication and controllable drug release application of the biomass microneedle patch [70]. (**B**). Design and fabrication of bioinspired oriented antibacterial sericin microneedles (OASM) for healing infected wounds. (a) Lamprey-teeth-inspired OASM was composed of sericin and zinc oxide nanoparticles (ZNPs), which would degrade and release ZNPs into the wounds. (b) The central short needles would insert into the wound site, and the edged long needles possessed a tilt angle that would provide a dragging force. (c) OASM would penetrate the skin tissues of the infected wound, the released ZNPs could kill the bacteria, and the edged titled needles would provide a dragging force to stimulate wound contraction. (d) Hair follicle regeneration and revascularization can be observed in the wound after OASM treatment. [71]. (**C**). (a) Schematic illustrations of the MN system loaded with ADSCs and PDGF-D. (b) Application of MNs for diabetic wound treatment [77]. (**D**). Schematic illustrations of the Fe-MSC-NVs/PDA MN patch for diabetic wound healing: (a) schematic of Fe-MSC-NVs/PDA MN patch; (b) schematic of the wound closure process [78]. Reproduced with permission from Bioactive Materials, Nano Letters, Advanced Functional Materials, and Advanced Science, respectively.

**Figure 5 polymers-15-04059-f005:**
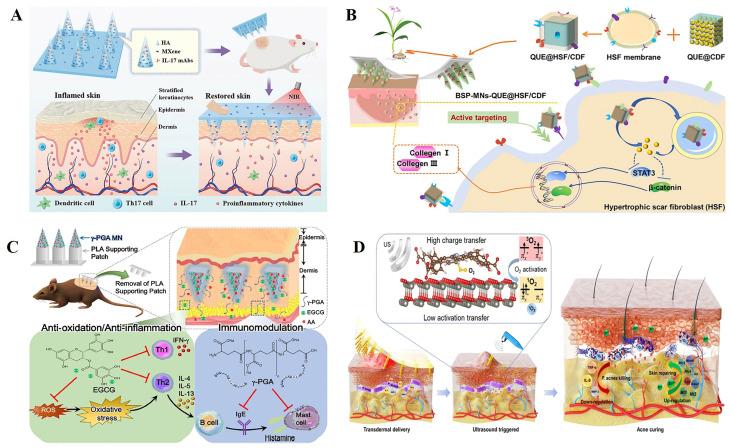
(**A**) Schematic design of the application of mAbs-loaded photothermal responsive MN patch for psoriasis treatment [135]. (**B**) Schematic illustration of fabrication and administration of BSP-MNs-QUE@HSF/CDF [136]. (**C**) Schematic illustration of the use of epigallocatechin gallate (EGCG)/L-ascorbic acid (AA)-loaded poly-γ-glutamate (γ -PGA) microneedles (MNs) to ameliorate AD-like symptoms in mice [137]. (**D**) Sonocatalytic mechanism and the treatment of acne through efficient sonodynamic ion therapy–based MN patch. US: ultrasound [138]. Reproduced with permission from Advanced Functional Materials, ACS Nano, Acta Biomaterialia, and Science Advances, respectively.

**Figure 6 polymers-15-04059-f006:**
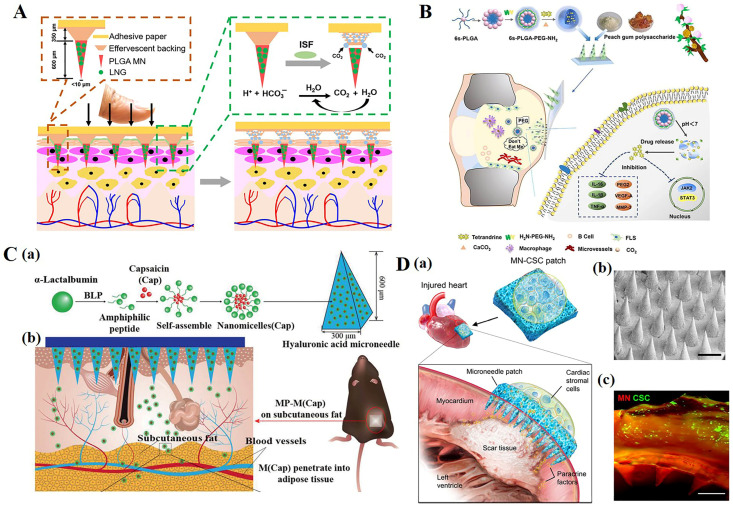
(**A**) Long-acting, reversible contraception using an effervescent microneedle patch [37]. (**B**) Drug-loaded multifunctional PLGA nanoparticles integrated with GP-fabricated dissolving microneedles for the local treatment of RA [144]. (**C**) Schematic illustration of a multifunctional microneedle patch with capsaicin-loaded micelles for suppressing adipogenesis and promoting adipocyte browning. (a) Micelles encapsulated with capsaicin (M (Cap)) are prepared from self-assembly of α-lac peptides and Cap; M (Cap) were then loaded into the microneedle patch made of hyaluronic acid (HA) and polyvinyl alcohol (PVA) with body temperature responsive melting property. (b) The microneedle patch (MP) encapsulated with M (Cap) was pressed onto the fur-removed skin of the left abdominal subcutaneous fat of high fat diet (HFD)-induced obese mice. [149] (**D**). Characterization of MN-CSCs. (a) Schematic showing the overall design used to test the therapeutic benefits of MN-CSCs for infarcted hearts. (b) SEM image of MN (scale bar: 500 μm). (c) Representative fluorescent image indicating that DiO-labeled CSCs (green) were encapsulated in fibrin gel and then integrated onto the top surface of the MN array (red) (scale bar: 500 μm) [161]. Reproduced with permission from Science Advances, Chemical Engineering Journal, Advanced Functional Materials, and Advanced Healthcare Materials, respectively.

## Data Availability

Not applicable.

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
