# Peer review of "The Progress in the Application of Dissolving Microneedles in Biomedicine"

_polymers, 2023, doi:10.3390/polym15204059_

Round 1

Reviewer 1 Report (Previous Reviewer 1)

1-     Could you clarify how microneedles (MN) are designed to penetrate the skin's cuticle without triggering pain-sensing neurons? What are the key properties of MN that make them suitable for direct access to the dermal Interstitial Fluid (ISF)?

2-     The text discusses two categories of MN-based medical sensing technologies: electrochemical biosensors and direct extraction of ISF. Could you provide specific examples of how each category is applied in medical diagnostics or monitoring?

3-     In the MN-based electrochemical biosensor, it's mentioned that the concentration of the target sample is converted into an electrical signal through the MN electrode. How is this conversion achieved, and can you provide an example of a specific target sample and its corresponding electrical signal?

4-     The text mentions a swelling MN patch made of methacrylic acid hyaluronic acid (MeHA) for ISF extraction. Could you explain how the hydrophilicity of MeHA contributes to efficient ISF extraction, and why additional equipment assistance is not required?

5-     The FAOM device for quantitative monitoring of blood glucose is described. Could you elaborate on how tubular DNA origami nanostructures and glucose oxidase molecules are integrated into the device's internal network for glucose monitoring?

6-     The insulin-loaded microneedle patch is discussed, highlighting the characteristics of glucose-responsive diffusion release. Could you explain how the presence of borate hydrogel enables this characteristic, and how crystalline silk fibroin components enhance skin permeability?

7-     In addition to blood glucose monitoring, the text mentions other applications of microneedles, such as detecting DNA, RNA, cytokines, exosomes, and small molecules. Can you provide an example of how microneedles are used for one of these applications?

8-     The microneedle patch for intelligent wound management is described. Could you explain how the microfluidic channel and porous ordered structure enable the analysis of various inflammatory factors, and how MXene electronic products contribute to motion monitoring?

9-     The use of drug-loaded biomimetic microneedle patches for promoting wound recovery is mentioned. Could you provide more details on how these patches work and the benefits they offer in wound healing?

            10- The authors can use the following article as a reference in the                                 introduction improvement: https://doi.org/10.3390/polym15092031

Author Response

Reviewer 2 Report (Previous Reviewer 3)

The review by Xueqing et al. was submitted after applying/obtaining copyright permission for the quoted images, which is ethically essential.

The authors mentioned references below these images, however, it is not sufficient to quote them. for each image, pease use the following sentence instead  "reproduced/adapted from [reference] with permission from [copyright owner]

Readable

Round 2

Reviewer 1 Report (Previous Reviewer 1)

All my comments are answered properly. The paper can be accepted in its current state.

This manuscript is a resubmission of an earlier submission. The following is a list of the peer review reports and author responses from that submission.

Round 1

Reviewer 1 Report

1-How have developments in nano and electrochemical technology expanded the applications of microneedles beyond transdermal drug delivery?

2-What specific biomedical fields other than drug delivery have benefited from the use of different types of microneedles?

3-Could you elaborate on the remarkable results achieved by dissolving microneedles in dermatological treatment, disease diagnosis, and monitoring, as well as vaccine delivery?

4-How can the advancements discussed in the abstract potentially lead to better solutions for clinical treatment in various biomedical applications?

5- The following article can be beneficial to use in the introduction discussion: https://doi.org/10.1080/17425247.2022.2119220

Reviewer 2 Report

  • Review was nicely designed and presented well. This review clearly explains the development of dissolving micro-needles and providing better solutions for clinical treatment.
  • Transdermal administration is clearly explained, detailed description provided in general and gradually going into specific details.
  • Each section is demonstrated with adequate details.
  • Font size and picture sizes increase is recommended as they are hard to follow, for instance in figure 2 and 4.
  • Appropriate references were cited.
  • Abbreviations elaboration in the introduction of a particular concept in the  article was thoughtful to remind the reader and walk them through. 
  • The conclusion part can be more specific with additional detailing with respect to the future directives. 
  • doi's are missing for the references 6 and 160

Reviewer 3 Report

Please find my comments in the attached file.

Not so adequate. 

Some typographic and punctuation errors need adjustments.

Round 2

Reviewer 3 Report

The authors didn't respond to the comments appropriately. 

Comment 1: They explained the contents of the previous reviews in the response letter, but not in the introduction of their current review. 

Comment 2: The figures are not original or have copyright permission.

Comment 3: Probably they didn't understand my comment. My comment means the authors should include a material and method section that should describe the keywords used to collect data for the review, the databases considered for their search on relevant articles, and the period covered (for example 2000-2023).

Comment  4: References as its 

Comment 5: Still there is minor TYPOS.

Comment 7: The authors misunderstood my comment.

My comment means that is there any published article by any of you in the area of microneedles? 

Minor typographic errors. 

Round 3

Reviewer 3 Report

Still, the authors didn't solve the problem of the figures. Most of the figures if not all are not original, thus the quality is too poor to be readable. 

Readable.